# Enhancement of Weldability at Laser Beam Welding of 22MnB5 by an Entrained Ultrasonic Wave Superposition

**DOI:** 10.3390/ma15144800

**Published:** 2022-07-09

**Authors:** Christian Wolf, Stephan Völkers, Igor Kryukov, Markus Graß, Niklas Sommer, Stefan Böhm, Maxim Wunder, Nadine Köhler, Peter Mäckel

**Affiliations:** 1Department for Cutting and Joining Manufacturing Processes, Institute for Production Technologies and Logistics, University of Kassel, Kurt-Wolters-Straße 3, 34125 Kassel, Germany; s.voelkers@uni-kassel.de (S.V.); i.kryukov@uni-kassel.de (I.K.); m.grass@uni-kassel.de (M.G.); n.sommer@uni-kassel.de (N.S.) s.boehm@uni-kassel.de (S.B.); 2isi-sys GmbH, Wasserweg 8, 34131 Kassel, Germany; mw@isi-sys.com (M.W.); nk@isi-sys.com (N.K.); pm@isi-sys.com (P.M.)

**Keywords:** laser welding, ultrasonic wave superposition, ultrasonic vibration, non-conventional welding, structural influence, grain refining, boron-alloyed heat treatable steel

## Abstract

In this paper, the potential of directional ultrasonic wave superposition by moving sound generators for laser beam welding of high-strength steel alloys 1.5528 (22MnB5) is studied. Steel sheets of identical thickness and in form of tailored blanks were joined in butt joint configuration. The influences of the various excitation parameters of the moving sound generators on the ultrasonic coupling and their influence on the distribution of the AlSi coating components within the melting zone and the weld seam characteristics are investigated. Etched cross-sections, scanning electron microscopy, energy dispersive X-ray spectroscopy, and electron backscattering measurements were used as the investigation methods to determine the AlSi distribution in the weld as well as its microstructure. The results presented a series of experiments which show that a suitable superposition of ultrasonic waves by the moving sound generators lead to a more homogeneous distribution of AlSi particles in the melt as well as to a finer microstructure within the weld, which improves the mechanical–technological properties.

## 1. Introduction

Due to the ongoing trend towards lightweight construction and the objective of reducing emissions and energy consumption, the use of high-strength steel materials in the field of mobility is increasing. Weight reduction becomes as significant as the potential of combining lightweight materials with one another as cost-effectively as possible and with little effort. Boron-alloyed quenched and tempered steels, such as the alloy 1.5528 (22MnB5), are increasingly used for these purposes. These steels are characterized by a comparatively very high hardness after the press hardening process. They are often joined in the form of tailored blanks with different sheet thicknesses using laser beam welding and, e.g., used in the automotive sector. The stability of the press hardening process is ensured by an AlSi coating system. However, this coating system causes substantial problems during welding as the AlSi alloy elements solidify within the weld metal of the joint. As a result of the impeded phase transformation, the hardness and strength properties of the weld are negatively affected [1,2,3,4,5,6,7].

In this regard, Khan et al. [8] showed a comparison of state-of-the-art approaches to solve the AlSi coating problem that occurs when welding of 22MnB5. Thereby, the strong decrease of the hardness and the strength properties within the weld due to the dilution of the fusion zone by an increased aluminum concentration is addressed as the main problem [8].

The formation of aluminum accumulations is particularly critical due to the formation of ferritic phases in these areas and a simultaneous suppression of martensitic phase transformation. Furthermore, it is stated that the formation of the ferritic phases and their approximately 40% lower hardness are the main causes of the reduction in the mechanical properties of the welded joint [1,8,9,10].

Consequently, Ganzer et al. [11] investigated the segregation of the AlSi coating components within the weld. The authors found that the AlSi coating components in the process of chain of welding and subsequent press hardening have a significant influence on the hardenability. Due to the accumulation of AlSi, the hardness of the weld seam is reduced by up to 50%, and the critical cooling rate to obtain a martensitic transformation is greatly increased [11].

In [12], Saha et al. also compared the individual failure mechanisms for different states of the material by tensile tests. They compared the failure mechanisms for the states of unaffected base metal (state 1), unaffected base metal followed by welding (same result as unaffected base metal), base metal after hot pressing (state 2), a welded joint followed by hot pressing (state 3), and a welded joint in which the base metal was first hot pressed and then welded (state 4). State 1 and state 2 fail within the unaffected base metal. For the condition where the unaffected base metal is welded, the failure also occurs within the base metal. The authors assume that the mechanical properties of the weld remain unaffected. For state 3, failure occurs in the region of the fusion boundary, which propagates along the α-ferritic network due to the low strength in the weld metal. A similar failure pattern is also observed for state 4, where failure along the fusion line can be explained by a comparatively high fraction of the ferritic phase [8,12,13].

Furthermore, Wenhu et al. [1] studied the influence of coating removal on the weldability. It was found that the welded joints with removed coatings exhibited significantly higher fracture elongation and ultimate tensile strength after hot pressing in contrast to the coated sheets. The observed reduction in elongation and strength is explained by the formation of d-ferrite [1].

However, there are solutions implemented industrially, such as removing the coating in the area of the joining zone or using filler material in wire or powder form, which intend to circumvent the problem of ferritic phase formation. However, these are not sufficiently cost-effective and continue to encounter technological difficulties. Furthermore, the underlying physics and chemistry of the proposed solutions have not been fully understood, so further research is still crucially needed [8].

Another viable method of solving the problem is to superimpose mechanically induced vibrations on the welding process. As such, Völkers et al. [14] investigated this method with stationary soundwave coupling and came to the conclusion that the introduced vibrations lead to a significantly more homogeneous distribution of aluminum and silicon within the weld metal. The stationary position of the mechanically induced oscillations, however, does not allow for a constant influence during the welding process since the distance between the weld pool and the soundwave coupling changes continuously.

A more efficient solution is to use a movable sound generation system, which is carried along at a defined distance and ensures a constant influence on the weld pool, e.g., as the one presented in the following paragraph. In this paper, the solution of the movable ultrasonic excitation will be presented and the influence of the movable system on the more homogeneous distribution of the AlSi constituents within the weld pool will be investigated. Furthermore, the influence of the sound superposition on the geometry of the weld and the formation of the grain size inside the weld will be investigated.

## 2. Materials and Methods

### 2.1. Material and Experimental Setup

In the following, the test setup and the test procedure for laser beam welding with moving ultrasonic wave superposition are presented. For laser beam welding, sheets of 22MnB5 (1.5528) with a thickness of 1.83 mm (including 150 µm AlSi layer thickness on each side) were used. The joining partners had a size of 300 × 150 mm^2^ and were joined over a joint length of 250 mm by means of an ytterbium fiber-laser (IPG YLS 10,000-S4, IPG Photonics GmbH, Burbach, Germany) with a nominal wavelength of 1070 nm and a focal diameter of 400 µm. During the experiments, the laser was focused onto the sheet surface and operated at a power of 3.5 kW. The traverse speed was set to 5 m/min. Figure 1 shows an overview of the test setup for the welding tests. A vacuum clamping table was used, with which various clamping situations can be reproduced. In addition to pure vacuum clamping, clamping situations with toggle clamps can also be simulated. For the welding tests, both joining partners were clamped with the aid of the aforementioned vacuum clamping table over 3 areas each with a size of 80 mm × 80 mm.

For the application of ultrasonic sound superposition, special moving sound generators (piezoshakers [15], shown in Figure 1) were developed by the company isi-sys GmbH (Kassel, Germany).

In the area of the weld zone, both joining edges can oscillate freely over a width of 5 mm each (see Figure 2). Before the welding tests, the plates were milled along the abutting edge to ensure a minimal gap. In addition, the sheets were laterally pretensioned against each other with vacuum before clamping. Furthermore, argon (purity 99.9996%) was used as inert gas as well as a crossjet to protect the lens in the laser optics from weld spatter. Additionally, the piezoshakers were held in place by a holding system, as depicted in Figure 1. The holder was attached to the laser optics so that it could be moved along the welding trajectory.

The piezoshakers can be laterally positioned or adjusted using locking slides to set the distance and angle of the piezoshakers to the weld pool and to compensate for any inclination of the laser optics. In addition, pneumatic cylinders were installed between the holder and the piezoshakers. On the one hand, these serve to press the piezoshakers onto the component surface with a defined force. On the other hand, they can compensate for certain component unevenness and thickness differences of the joining partners (e.g., in the case of tailored blanks).

### 2.2. Visualization of the Vibration Coupling into the Joining Partners

For the welding test series, a piezoshaker system based on a special high-frequency amplifier combined with a piezoshaker was developed in cooperation with the company isi-sys GmbH (Kassel, Germany) in several optimization steps. The piezoshaker can be equipped with different piezo actuators and contact pins for different component geometries and was investigated with regard to vibration coupling and sliding properties. For this purpose, full-field vibration measurements by time-average shearography (SE2-5.04, isi-sys GmbH, Kassel, Germany) were used in order to monitor the stationary state for different excitation parameters as well as the positioning of the piezoshaker with reference to the weld. An exemplary measurement is shown in Figure 3. It was carried out on the test setup for the welding tests on two positioned but unwelded sheets. Shearography can be used to visualize the standing waves by time average or the transverse waves in the joining partners. For this purpose, both piezoshakers were pressed onto the component surface by means of pneumatic cylinders and excited at an exemplary frequency. The dark areas of the image in Figure 3 represent the transverse soundwaves in the component. It can be seen very clearly that both joining partners oscillate along the joining zone.

First of all, investigations regarding the soundwave coupling and the proof of the ultrasonic influence on the weld seam were carried out on welded joints with a plate thickness of 1.83 mm. Figure 4 shows an exemplary shearography measurements that can be used to visualize the influence of two parameters on the soundwave coupling. In the two top figures, a comparison between a low and higher amplitude (gain factor of the sinusoidal function) is shown. The bottom figures visualize a comparison of the influence of two different contact pressure values of the piezoshaker on the component surface. For either case, increasing the amplitude or increasing the contact pressure leads to a more pronounced component vibration. From a technical point of view, however, a compromise is necessary for the contact pressure and vibration amplitude with regard to vibration transmission and the sliding motion of the piezoshaker head. Nonetheless, all four examples exhibit a clear wavelike fluttering of the joining zone. Here, the dark areas reflect high inclinations in the vertical direction (dW/dy) of the surface.

### 2.3. Specimen Characterization

The specimen surface was characterized by means of an optical macroscope (Leica Z16) and white light interferometry (WLI, FRT GmbH, Bergisch Gladbach, Germany). Following the welding experiments, cross-sections were extracted perpendicular to the welding direction using electric discharge machining. Subsequently, these cross-sections were mechanically ground using silicon-carbide paper, polished and etched using a solution of 3% HNO_3_ and 97% ethanol. Inspection using light microscopy was carried out with a maximum magnification of 100× (Leica DM2700, Leica Germany GmbH, Wetzlar, Germany). Moreover, the elemental distribution and phase formation was investigated using energy dispersive X-ray spectroscopy (EDS, Bruker XFlash 6160, Bruker Nano Analytics, Berlin, Germany) and electron backscatter diffraction (EBSD, Bruker e^—^flash, Bruker Corporation, Billerica, USA), respectively. The analysis was carried out using a scanning electron microscope (SEM, Zeiss REM Ultra Plus, Carl Zeiss Microscopy Deutschland GmbH, Oberkochen, Germany) using an acceleration voltage of 20 kV. The EBSD-analysis was performed with a resolution of 250 nm per pixel Prior, and the mechanically ground samples were vibration-polished using colloidal silicone suspension (Struers OP-S NonDry, Struers GmbH, Willich, Germany) with a grit size of 0.25 µm.

## 3. Results and Discussion

### 3.1. Influence of the Sound Generator on the Component Surface

The contact area of the piezoshakers with the sheets was examined using optical macroscopy and WLI, as described beforehand. This is important in order to evaluate possible damage to the sheet surface caused by the contact surfaces of the piezoshakers moving on the component surface. The contact pressure of the piezoshaker with the component surface while it is moving along in the welding process creates a slight trace on the component surface, as the left image in Figure 5 visualizes. A trace of the piezoshaker on the component surface is examined using a macroscope. A non-continuous, slightly shiny impression can be seen in this area. The surface topography of the track was measured with the aid of WLI in a measuring area with a size of 10 mm × 10 mm. A 3-dimensional representation of the measured surface topography is shown in Figure 5b and a 2-dimensional representation is shown in Figure 5c. It can be seen that in the area of the contact surface with the piezoshaker, only the roughness peaks in the range of approx. 1.5 µm are smoothened or polished. Therefore, it is assumed that this influence on the surface has a negligible effect on the corrosion resistance or a subsequent coating.

### 3.2. Influence of Ultrasound on the Mixing of the Weld Pool and the Grain Structure of Joint Partners with the Same Wall Thickness

A comparison between a weld without ultrasonic wave superposition and a weld with ultrasonic wave superposition is given in Figure 6a,b, respectively. The top two figures show etched cross-sections of the weld. In cross-section Figure 6a, it can be seen that the AlSi coating, visible as light areas within the weld, is very roughly segregated within the fusion zone. In the right cross-section, it can be seen that the AlSi coating is finely distributed. Figure 6c,d depict the results of the EDS-analysis. The aluminum distribution is shown as a false-color image with a scale ranging from 0–20 wt.%. In both EDS measurements, a partial accumulation of about 4 wt.% Al is present within the weld. However, as illustrated by Figure 6c, it can be seen that the aluminum of the AlSi coating is precipitated much more coarsely within the weld, which has a negative effect on the strength of the weld and was already shown in [9]. In contrast, the lower right figure shows a finer and more homogeneous distribution within the weld.

Furthermore, electron backscatter diffraction (EBSD) measurements were performed on selected samples. Figure 7 shows the superimposed SEM images and sectioned inverse pole figure mappings (IPFM). The cross-section in Figure 7a was obtained by laser beam welding without ultrasonic wave superposition, while the weldment in Figure 7b was performed with ultrasonic wave superposition. The respective section of the EBSD measurement shows the IPFM with grain orientations plotted perpendicular to the traverse direction and sheet surface. As the measurements illustrate, the base metal and the heat-affected zone exhibit a very fine-grained microstructure. In the weld without ultrasonic wave superposition, a coarse-grained microstructure as a result of substantial columnar grain growth is found. This microstructure forms predominantly in the case of directional heat dissipation perpendicular to the temperature gradient, which is furthered by the high cooling rates of laser beam welding. In the section of the EBSD measurement with ultrasonic wave superposition, the sound-induced dendrite shearing during the solidification process results in a significantly refined microstructure, which may have a very positive effect on the strength and toughness of the weld, as demonstrated in [8]. This effect can already be seen through the reflection pattern which is visualized gray in the background. In the weld without ultrasonic wave superposition, a clear transition between the weld metal and heat-affected zone can be distinguished. In the weld with ultrasonic wave superposition, no defined boundary can be seen.

### 3.3. Application of Ultrasound to Complex Geometries

In the following, the transfer of the results from the ultrasonic wave superposed welding tests with identical sheet thickness to more complex geometries is described using the example of tailored blanks. The experimental setup used to perform the welding tests with the tailored blanks was equivalent to the one described beforehand. However, the welding optics needed to be tilted by 10° parallel to the welding trajectory, yielding a slightly ellipsoidal beam focus. Figure 8 shows the welded joint of such tailored blanks. For this purpose, two plates (22MnB5) with different plate thicknesses were butt-welded. The plate on the left has a thickness of 1.25 mm while the plate on the right features a thickness of 1.83 mm, each including 150 µm AlSi coating thickness. The welding process with ultrasonic wave superposition was performed with the piezoshakers superimposed with soundwaves at a defined distance. The left image shows the top side of the weld, the middle image shows a cross-section without ultrasonic wave superposition, and the right image shows a cross-section with ultrasonic wave superposition. In the micrographs (cf. Figure 8b,c), the AlSi coating can be seen as a white area on the upper and lower edges. In the cross-section without ultrasonic wave superposition, the accumulations of the AlSi coating within the weld seam can be distinguished as bright areas, corresponding to the welding experiments with identical sheet thickness. In the area of the weld seam with ultrasonic wave superposition, the coating components are finely distributed, so that no pronounced accumulations of the AlSi coating have occurred.

Furthermore, Figure 8b shows that in the cross-section without ultrasonic wave superposition, a clear demarcation between the heat-affected zone and the weld seam can be seen due to the transition of the grain structure. In Figure 8c, on the other hand, no clear transition can be seen, which can be attributed to the refined grain structure and more homogeneous elemental distribution. Figure 9 depicts the corresponding EDS measurements for aluminum as false-color image. It should be noted that the cross-section specimens were re-polished for preparation of the EDS measurement and, thus, the investigated microscopy may have slightly shifted. The scale bar ranges from 0–9 wt.%. Outside of the weld, an alloy component of the element aluminum greater than 9 wt.% can be seen on the top and bottom edges of the sheet within the coating. Along the upper and lower edges of the weld, an accumulation of aluminum is also present. In the welded joint without ultrasonic wave superposition (cf. Figure 9a), a high content is also present within the weld, as was already observed in the cross-section in Figure 6c. In the welded joint with ultrasonic wave superposition (cf. Figure 9b), an accumulation of the alloying element aluminum is also present on the top and bottom of the weld, but at a much lower level of about 5–6 wt.%. Within the weld metal, a more homogeneous aluminum-distribution can be observed. Furthermore, for the sample with ultrasonic wave superposition, a clear separation between the weld seam and the base material can be seen in the EDS-measurement, since the aluminum alloy components are distributed up to the fusion line but do not reach the unmelted base material.

Figure 10 depicts an EBSD measurement of the corresponding tailored blank welded joint with ultrasonic superposition from Figure 8c. The image in the upper left corner shows an overview of the weld from the scanning electron microscope at 30× magnification with a marking for the measurement region. It can be derived from the detailed quality map (IQM) and IPFM that a typical martensitic phase structure is present due to the rapid cooling rates that occur during laser beam welding. The phase fraction of martensite is 97.7% and that of austenite is 2.32% in the investigated area. Furthermore, the convergence of the solidification fronts within the investigated area of the weld can be detected. Notably, a clear boundary can be seen in this area. The IPFM on the right-hand side of Figure 10 illustrates that—analogue to the experiments performed with an identical sheet thickness (cf. Figure 7)—soundwave superposition leads to a fine-grained microstructure. Furthermore, it can be seen that the grain growth is predominantly parallel to the direction of heat dissipation. In the area where the solidification fronts meet, a local change in the predominant grain orientation can be seen inside the weld. Here, the grain orientation is perpendicular to the heat flow direction, and can be seen in the EBSD measurement as well as in the image on the left.

In order to further investigate the influence of excitation parameters on the solidification microstructure, weldments were performed with decreased soundwave excitation frequencies. Figure 11a shows a detailed IQM of a tailored blank weld metal jointed with soundwave superposition with lower frequencies (@10 kHz). Despite the soundwave superposition, the microstructure still features areas of increased Al concentration. It can be seen that a fine-grained martensitic microstructure is predominant. In contrast, there are areas where the phase transformation is suppressed due to an elevated Al concentration, leaving a ferritic phase. Such a ferritic phase agglomeration is highlighted in Figure 11b. Moreover, the IPFM of the region in Figure 11b visualizes that the ferritic grain solidifies with different orientation directions, analogue to the martensitic phase. Since the ferritic phase has a drastically lower strength, crack growth occurs along these areas in the event of damage (cf. [8,10]).

## 4. Conclusions

The paper presented the potential of directional ultrasonic wave superposition by entrained ultrasound generators during laser beam welding of a coated, high-strength steel alloys. The individual influences of the excitation parameters of the piezoshakers on the soundwave coupling was investigated. Furthermore, the influence of the ultrasonic wave superposition on the microstructure formation and the distribution of the AlSi coating constituents within the weld seam and their expression were studied. Shearography measurements, etched cross-sections as well as EBSD- and EDS-measurements were performed for an in-depth analysis of the influence of soundwave coupling.

Based on the results, it can be concluded that an influence of soundwave superposition on the welded joint with a moving system could first be demonstrated on a butt-joint with identical wall thickness. It could be demonstrated that ultrasonic wave superposition during laser beam welding of 1.5528 (22MnB5) effectively prevents the accumulation of the AlSi coating in the weld, which is indispensable for a subsequent forming process. This could be verified by etched cross-sections as well as EDS- and EBSD-analysis. Subsequently, the findings were transferred to the application of soundwave superimposed laser beam welding of tailored blanks. The positive influence of ultrasonic wave superposition identified beforehand could also be demonstrated in this case. It is assumed that the propagation of ultrasound waves within the melt results in stronger weld pool convection. The stronger circulation of the melt during the solidification process leads to dendrite shearing, resulting in a finer-grained microstructure. EBSD measurements support this assumption on welded joints with the same wall thickness as well as on tailored blanks. Furthermore, it is found that an excitation with lower frequencies of around 10 kHz has no sufficient influence. The development of a moving piezoshaker system enables a continuous and even intensity distribution along the weld seam, allowing for high welding speeds with ultrasonic wave superposition. This eliminates the need for time-consuming seam preparation or the use of filler material for laser welding of coated, high-strength steel alloys.

## 5. Patents

A patent was applied for with the title “Verfahren und Vorrichtung zur gezielten Anregung der Schweißschmelze mit Schwingungen und Schallwellen durch relatives Mitbewegen von Schwingungserregern mit der Schweißzone”. The file number for the patent is 10 2021 003 035.4.

## Figures and Tables

**Figure 1 materials-15-04800-f001:**
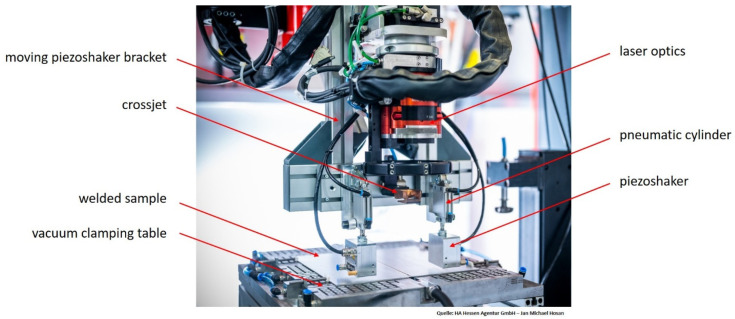
Photographic image depicting an overview of the experimental setup for welding experiments.

**Figure 2 materials-15-04800-f002:**
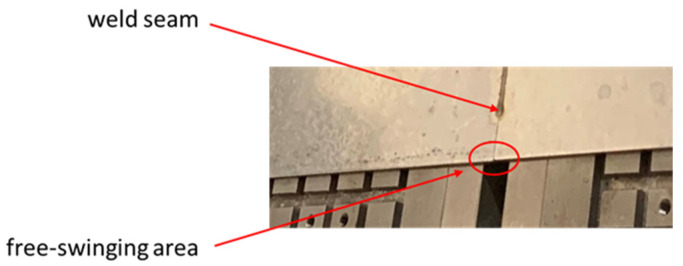
Detailed view of the joining edge and free-swinging area.

**Figure 3 materials-15-04800-f003:**
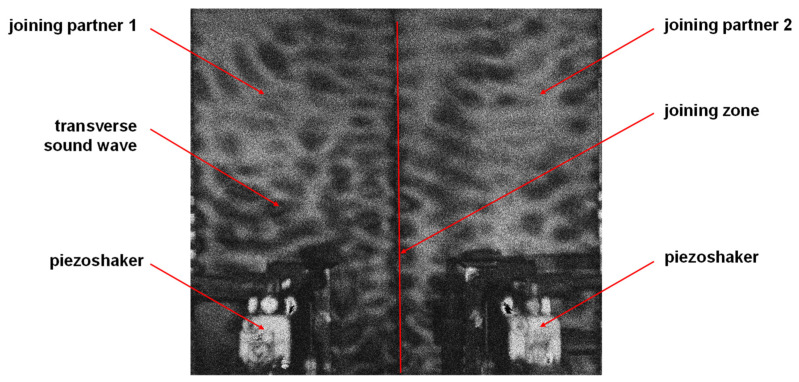
Vibration visualization by means of shearography.

**Figure 4 materials-15-04800-f004:**
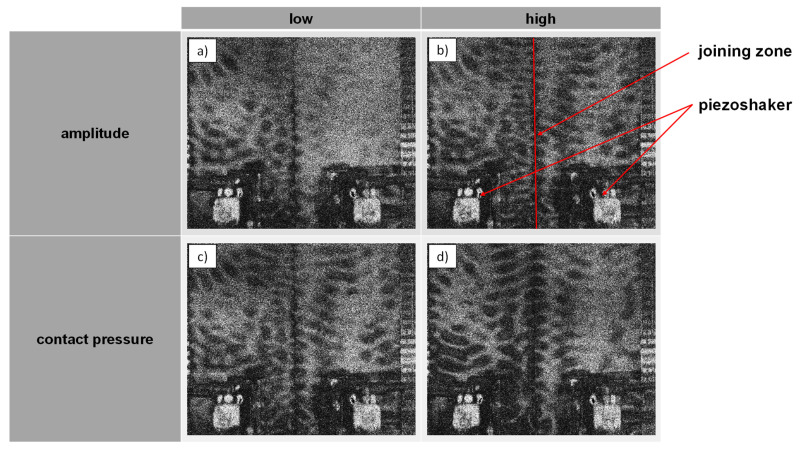
Shearography measurement to investigate the influence of excitation amplitude (**a**,**b**) and contact pressure (**c**,**d**) on the soundwave coupling.

**Figure 5 materials-15-04800-f005:**
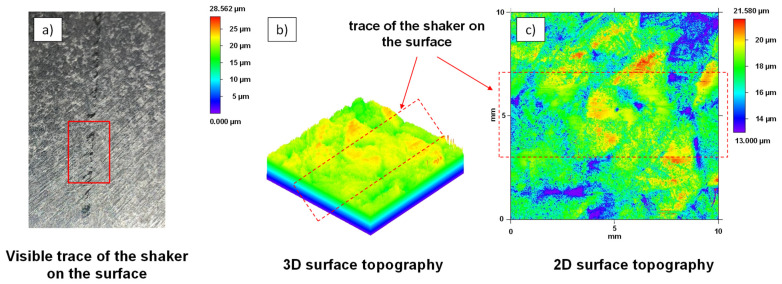
(**a**) Macroscopy overview of the sheet surface to inspect the piezoshaker-trace and (**b**,**c**) surface topography measurement of the sheet surface marked in (**a**) using WLI.

**Figure 6 materials-15-04800-f006:**
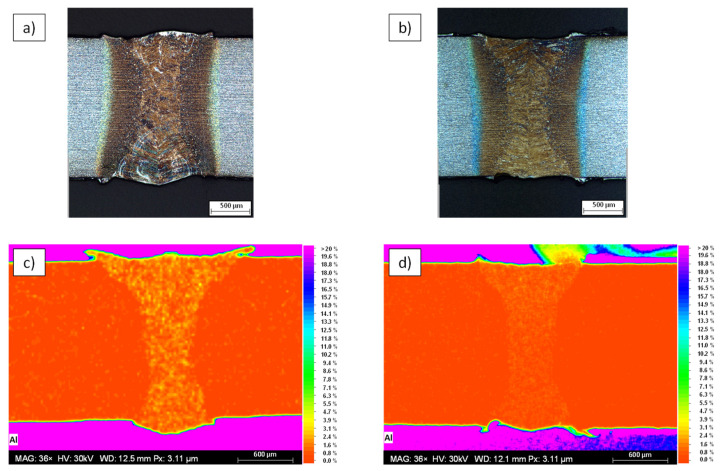
Light microscopy images depicting etched cross-sections of the weld obtained without (**a**) and with (**b**) ultrasonic excitation. (**c**,**d**) illustrate the corresponding EDS-maps for aluminum.

**Figure 7 materials-15-04800-f007:**
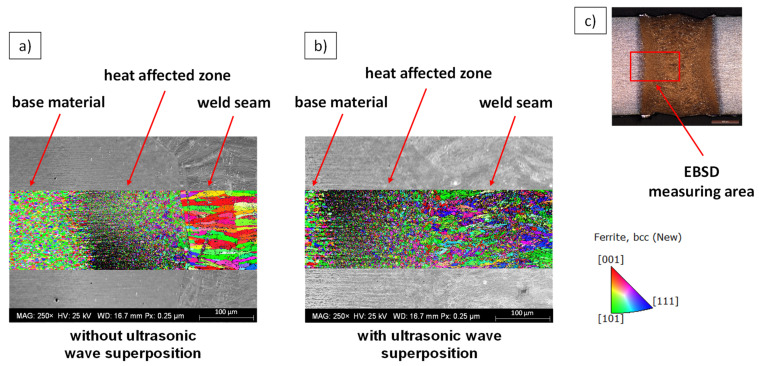
Superimposed SEM images and IPFM to study the influence of superposition without (**a**) and with (**b**) ultrasonic waves on the solidification mechanisms of 22MnB5. Grain orientations plotted perpendicular to the traverse direction and sheet surface. Overview of the EBSD measuring area within the cross-section (**c**).

**Figure 8 materials-15-04800-f008:**
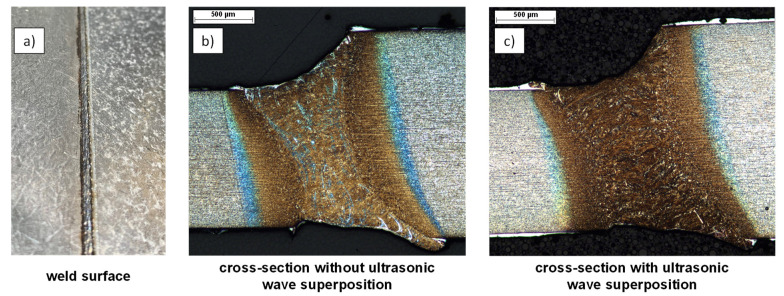
(**a**) Photographic image depicting the surface of laser-welded tailored blanks. (**b**,**c**) show etched cross-sections to investigate the influence of ultrasonic wave superposition on the distribution of AlSi within the weld metal.

**Figure 9 materials-15-04800-f009:**
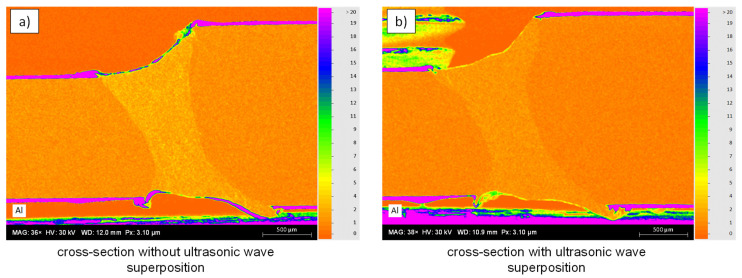
Comparison of EDS-analysis for aluminum in a false color image (0–9 wt.%) on welded tailored blanks. (**a**) depicts the cross-section obtained by welding without ultrasonic superposition and (**b**) with ultrasonic superposition.

**Figure 10 materials-15-04800-f010:**
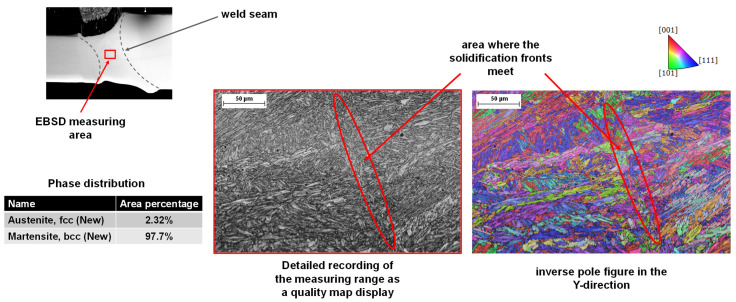
SEM image depicting the area for detailed EBSD-analysis of the microstructural evolution of tailored blanks welded wit ultrasonic soundwave superposition. Detailed EBSD-analysis obtained at a magnification of 400×. Grain orientations plotted parallel to the traverse direction.

**Figure 11 materials-15-04800-f011:**
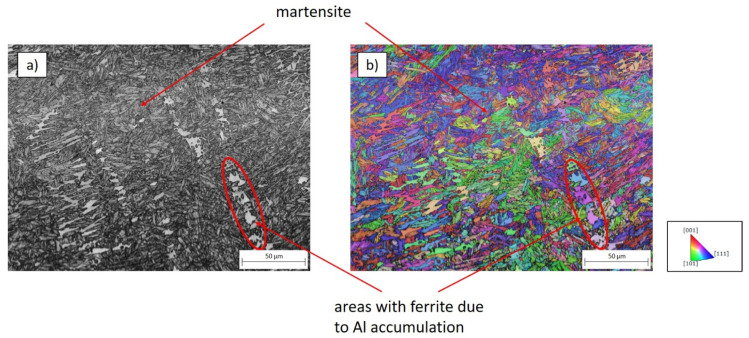
Comparison of the phase formation within the weld metal of tailored blanks, jointed with soundwave superposition at lower frequencies. (**a**) IQM and (**b**) IPFM. Grain orientations plotted parallel to the traverse direction.

## Data Availability

The data required for these results cannot be not be disclosed at this time because the data are part of an ongoing investigation.

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
