# Peer review of "Enhancement of Weldability at Laser Beam Welding of 22MnB5 by an Entrained Ultrasonic Wave Superposition"

_materials, 2022, doi:10.3390/ma15144800_

Round 1

Reviewer 1 Report

1. Please change the first sentence of Introduction. The article is about welding steel, not light metal alloys.

2. line 114: please remove the editorial artifacts and insert the correct reference!!!

3. At the end of the Introduction section, please enter the aim of the article.

4. line 193: please remove the editorial artifacts and insert the correct reference!!!

5. line 208: please remove the editorial artifacts and insert the correct reference!!!

6. line 223: please remove the editorial artifacts and insert the correct reference!!!

7. please mark Figure 7 as (a-d) and describe its components. Please enlarge the section on ferrite.

8. line 270: please remove the editorial artifacts and insert the correct reference!!!

9. line 283: as above.

10. line 305: as above

Author Response

Dear Sir or Madam,
Thank you for your valuable feedback and comments on our submitted paper. We regret if we have not answered all relevant questions within the scope of the paper to your satisfaction. We have therefore made changes to the manuscript in accordance with the reviewer comments. Below is a list of the responses to your comments.

Point 1: Please change the first sentence of Introduction. The article is about welding steel, not light metal alloys.

Response 1: The first sentence of the introduction was adjusted

Point 2: line 114: please remove the editorial artifacts and insert the correct reference!!!

Response 2: was adjusted

Point 3: At the end of the Introduction section, please enter the aim of the article.

Response 3: was adjusted

Point 4: line 193: please remove the editorial artifacts and insert the correct reference!!!

Response 4: was adjusted

Point 5: line 208: please remove the editorial artifacts and insert the correct reference!!!

Response 5: was adjusted

Point 6: line 223: please remove the editorial artifacts and insert the correct reference!!!

Response 6: was adjusted

Point 7: please mark Figure 7 as (a-d) and describe its components. Please enlarge the section on ferrite

Response 7: was adjusted in the picture as well as in the text

Point 8: line 270: please remove the editorial artifacts and insert the correct reference!!!

Response 8: was adjusted

Point 9: line 283: as above.

Response 9: was adjusted

Point 10: line 305: as above

Response 10: was adjusted

We sincerely hope to have addresses all remaining inconsistencies and look forward to hearing your feedback.

Best regards,

Christian Wolf

Reviewer 2 Report

1.  In the abstract, please do focus on the methodology of investigation and the main findings instead of the presentation of the subject.

2. The authors declared that “Moreover, the IPFM of the region in Figure 11 (b) visualizes that the ferritic grain solidify with different orientation directions, analogue to the martensitic phase. Since the ferritic phase has a drastically lower strength, crack growth occurs along these areas in the event of damage (see Chapter 1).”. What is the “see Chapter 1”?

3.The fine-grained martensitic microstructure is predominant within the weld metal of tailored-blanks, jointed with sound wave superposition at lower frequencies. Why?

4.  One suggestion, “cross section” and “cross-section” was used in this paper. Which is right?

5.  The Conclusion should be more informative the experimental data.

Author Response

Dear Sir or Madam,
Thank you for your valuable feedback and comments on our submitted paper. We regret if we have not answered all relevant questions within the scope of the paper to your satisfaction. We have therefore made changes to the manuscript in accordance with the reviewer comments. Below is a list of the responses to your comments.

Point 1: In the abstract, please do focus on the methodology of investigation and the main findings instead of the presentation of the subject.

Response 1: the abstract was revised

Point 2: The authors declared that “Moreover, the IPFM of the region in Figure 11 (b) visualizes that the ferritic grain solidify with different orientation directions, analogue to the martensitic phase. Since the ferritic phase has a drastically lower strength, crack growth occurs along these areas in the event of damage (see Chapter 1).”. What is the “see Chapter 1”?

Response 2: precise sources were given

Point 3: The fine-grained martensitic microstructure is predominant within the weld metal of tailored-blanks, jointed with sound wave superposition at lower frequencies. Why?

Response 3: Circulating and grain refining effects already occur due to the sound superposition with low frequencies during the cooling process. However, the energy introduced is not yet sufficient to prevent all accumulation of the AlSi coating. The austenitic microstructure within the weld folds over into the martensitic microstructure due to the high cooling rates during cooling. The critical cooling rate for generating the martensite is approx. > 20 K/s, which is clearly exceeded.

Point 4: One suggestion, “cross section” and “cross-section” was used in this paper. Which is right?

Response 4: it was changed to a unified term with "cross-section"

Point 5: The Conclusion should be more informative the experimental data.

Response 5: The authors think that the conclusion is Informative presented and summarizes all the important results of the experimental data.

We sincerely hope to have addresses all remaining inconsistencies and look forward to hearing your feedback.

Best regards,

Christian Wolf

Round 2

Reviewer 2 Report

No